# Phenotypic Validation of the Cotton Fiber Length QTL, *qFL*-*Chr*.25, and Its Impact on AFIS Fiber Quality

**DOI:** 10.3390/plants14131937

**Published:** 2025-06-24

**Authors:** Samantha J. Wan, Sameer Khanal, Nino Brown, Pawan Kumar, Dalton M. West, Edward Lubbers, Neha Kothari, Donald Jones, Lori L. Hinze, Joshua A. Udall, William C. Bridges, Christopher Delhom, Andrew H. Paterson, Peng W. Chee

**Affiliations:** 1Department of Crop and Soil Sciences and Institute of Plant Breeding, Genetics, and Genomics, University of Georgia-Tifton Campus, Tifton, GA 31793, USA; sameer@uga.edu (S.K.); inbrown@uga.edu (N.B.); pawanks76@gmail.com (P.K.); dawest@uga.edu (D.M.W.); elubbers@uga.edu (E.L.); 2Bayer CropScience, 700 W Chesterfield Pkwy W., Chesterfield, MO 63017, USA; 3Office of National Programs, USDA-ARS, Beltsville, MD 20705, USA; neha.kothari@usda.gov; 4Cotton Incorporated, Cary, NC 27513, USA; djones@cottoninc.com; 5Crop Germplasm Research Unit, USDA-ARS, College Station, TX 77845, USA; lori.hinze@usda.gov (L.L.H.); joshua.udall@usda.gov (J.A.U.); 6School of Mathematical and Statistical Sciences, Clemson University, Clemson, SC 29631, USA; wbrdgs@clemson.edu; 7Cotton Structure and Quality Research Unit, Southern Regional Research Center, USDA-ARS, New Orleans, LA 70124, USA; chris.delhom@usda.gov; 8Plant Genome Mapping Lab., University of Georgia, Athens, GA 30602, USA; paterson@uga.edu

**Keywords:** upland cotton, interspecific introgression, near-isogenic introgression lines, fiber quality, quantitative trait locus

## Abstract

Advances in spinning technology have increased the demand for upland cotton (*Gossypium hirsutum* L.) cultivars with superior fiber quality. However, progress in breeding for traits such as fiber length is constrained by limited phenotypic and genetic diversity within upland cotton. Introgression from *Gossypium barbadense*, a closely related species known for its superior fiber traits, offers a promising strategy. Sealand 883 is an obsolete upland germplasm developed through *G. barbadense* introgression and is known for its long and fine fibers. Previous studies have identified a fiber length quantitative trait locus (QTL) on Chromosome 25, designated *qFL*-*Chr*.25, in Sealand 883, conferred by an allele introgressed from *G. barbadense*. This study evaluated the effect of *qFL*-*Chr*.25 in near-isogenic introgression lines (NIILs) using Advanced Fiber Information System (AFIS) measurements. Across four genetic backgrounds, NIILs carrying *qFL*-*Chr*.25 consistently exhibited longer fibers, as reflected in multiple length parameters, including UHML, L(n), L(w), UQL(w), and L5%. Newly developed TaqMan SNP diagnostic markers flanking the QTL enable automated, reproducible, and scalable screening of large populations typical in commercial breeding programs. These markers will facilitate the incorporation of *qFL*-*Chr*.25 into commercial breeding pipelines, accelerating fiber quality improvement and enhancing the competitiveness of cotton against synthetic fibers.

## 1. Introduction

Upland cotton (*Gossypium hirsutum* L.) is the world’s leading fiber crop for lint production, supplying the majority of natural fibers for the textile industry. With advancements in spinning technology and increasing consumer demand for high-performance fabrics, the need for cultivars with superior fiber quality has intensified. Textile mills particularly value fibers that are long, fine, strong, and uniform, as these traits enhance spinning efficiency, reduce processing waste, and improve the quality of the end-use product [1]. High-quality cotton fibers enable the production of stronger and finer yarns, which are crucial for manufacturing premium textiles, including high-thread-count fabrics, performance wear, and luxury apparel. As synthetic fibers continue to challenge cotton’s market share, continued improvements in fiber quality are critical to ensuring that upland cotton remains the preferred natural fiber in the global textile industry.

Cotton researchers face numerous challenges when seeking to improve fiber quality. First, a negative correlation exists between fiber quality and yield-related traits [2], leading breeders to prioritize yield while applying less stringent selection for fiber quality traits in breeding new cultivars. Second, fiber quality traits are quantitatively inherited and controlled by many genetic loci [1], making their expression highly influenced by environmental factors [3]. Third, these genetic loci can interact in complex ways depending on the host’s genetic background [4]. Lastly, the upland cotton gene pool has very low genetic diversity [5]. This lack of diversity, exacerbated by decades of selection for fiber quality traits [6,7], suggests that beneficial mutations with large effects on fiber traits may already be fixed within cultivated germplasm [8].

Introgression of beneficial alleles for fiber quality from the secondary gene pool of *G. hirsutum* offers the quintessential solution for expanding upland cotton’s genetic diversity by introducing novel alleles for fiber quality traits [9,10,11,12]. *Gossypium barbadense*, commonly referred to as Extra-Long Staple, Sea Island, Egyptian, or Pima cotton, is a key source for fiber quality alleles due to its naturally superior fiber properties [11]. Numerous documented accounts of the mapping of quantitative trait loci (QTL) exist for fiber quality in interspecific populations involving these two species [13,14,15,16,17,18,19,20,21,22,23,24]. The introgression of favorable fiber quality alleles from *G. barbadense* into predominantly *G. hirsutum* genetic backgrounds has been well documented [13,17,23,25,26,27,28,29]. Although fiber quality traits are quantitatively inherited, fiber length, strength, and fineness exhibit higher heritability than yield, making these traits more responsive to selection [4].

Although researchers have identified several fiber length QTLs, only a few have been able to validate these QTLs [22,30,31,32]. Additionally, much remains unknown about how these QTLs may interact in different genetic backgrounds, potentially leading to unintended outcomes. For example, a QTL may exhibit variable expression depending on the environmental conditions, positively or negatively affecting other traits, or may fail to express entirely in specific genetic backgrounds. This complexity underscores the need to test QTL deployment across multiple genetic backgrounds to better understand these interactions before incorporating them into breeding programs [30] or utilizing them for cloning of fiber-related genes [33].

Kumar et al. [17] showed that “Sealand 883”, an obsolete cultivar from the Pee Dee Breeding Program, developed via interspecific hybridization [34], produced significantly longer and finer fibers than its recurrent upland parent, “Coker Wilds”. They suggest stable introgression of *G. barbadense* genes from the Sea Island donor parent, “Bleak Hall”. Genetic analysis revealed that Sealand 883 contained at least 10 stable introgressions from *G. barbadense* on five chromosomes (5, 11, 15, 16, and 25), ranging from 8 to 39 cM in size, collectively spanning a total of 235 cM, or 4% of the *G. hirsutum* genome. QTL analysis identified a key introgression on Chromosome 25 associated with fiber length (*qFL*-*Chr*.25), fiber strength (*qSTR*-*Chr*.25), and micronaire (*qMIC*-*Chr*.25). This region, approximately ~7 cm in recombination size and spanning 2.1 Mb, was flanked by SSR markers BNL827 and NAU2714. Association analysis indicated that *qFL*-*Chr*.25 explained 4.4% to 9.4% of the fiber length variation in the original mapping population [17].

The introgression of *qFL*-*Chr*.25 into four upland cotton genetic backgrounds representing major U.S. cotton production regions—Acala SJ4 (Southwest), Paymaster HS26 (Texas High Plains), Deltapine 50 (Mississippi Delta), and GA 2004089 (Southeast)—allowed for the assessment of the QTLs phenotypic effects [13]. Except for the Georgia background, these cultivars were bred in the early 20th century for regional adaptation [35,36]. Their diverse pedigree and significant fiber length variation made them ideal for evaluating the genetic effects of *qFL*-*Chr*.25. Brown et al. [13] crossed Sealand 883 with each background, and the *F*_3_ bulk sister lines (BSLs) were selected using marker-assisted selection for QTL(+) and QTL(−) lines. Brown et al. [13] then advanced these near-isogenic lines to *F*_6_ and *F*_7_ generations and evaluated them in field trials in 2014 and 2015. While all QTL(+) lines exhibited numerical differences, only three backgrounds showed statistically significant effects [30]. Additionally, Brown et al. [13] refined the map position of *qFL*-*Chr*.25 from a ~2.1 Mb region to ~800 kb region, enriching it with additional SSR markers, UGT2504 and UGT2509.

The present study seeks to further investigate the effect of *qFL*-*Chr*.25 using a more advanced set of near-isogenic introgression lines (NIILs) from common genetic lineages, combined with fiber quality testing through the Advanced Fiber Information System (AFIS). High Volume Instrument (HVI) fiber testing, previously utilized by Brown et al. [13,30], differs significantly from AFIS fiber testing. HVI measures a subsample of the fibers using an optical scanner to determine fiber length; in contrast, AFIS employs an aeromechanical separator to isolate individual fibers, measuring their properties—including length and maturity—using infrared beams and electro-optical sensors [37]. Consequently, AFIS provides unique fiber data, including a detailed length distribution based on individual fibers, offering more profound insights into the profile of individual samples. In contrast, HVI reports only the upper half mean length (UHML), which represents the average length of the longer half of fibers, potentially misrepresenting the true length by excluding short fibers. Additionally, while HVI micronaire (MIC) values combine fiber fineness and maturity into a single reading, AFIS measures fiber width separately. AFIS also independently evaluates maturity traits, such as the maturity ratio, which is calculated from circularity (Theta), and immature fiber content, based on the percentage of fibers with Theta < 0.25 [38]. By separating these measurements, AFIS provides a more precise and comprehensive fiber quality profile.

## 2. Results

### 2.1. Analysis of Variance (ANOVA)

TaqMan SNP genotyping markers (Table 1) enabled the identification of *G. barbadense*-specific alleles, allowing for the prescreening of material prior to field evaluations. The twenty-one genotypes in this study were tested in eight different environments: Tifton, GA, from 2020 to 2023; College Station, TX, and Plains, GA, in 2021 and 2022. Field trials were conducted in a randomized complete block design with four replications each year, except in 2020 and 2023, which had three replications. While fiber length was the primary trait of interest, analyses of variance showed significant differences among main effects for all the measured fiber traits in both HVI and AFIS (Table 2 and Table 3). Notably, there were no significant differences among replications, indicating consistency across trials. Significant differences were observed for both environment and genotype main effects across all traits, with varying genotype by environment interaction. While some traits showed significant genotype by environment interactions when data were combined across genetic backgrounds, this significance was not consistently observed when analyzed within individual backgrounds.

### 2.2. Correlation Among Fiber Traits

Pearson correlation coefficients among fiber quality traits were calculated across all eight environments (Table 4), revealing numerous significant relationships. Comparing HVI fiber properties with AFIS measurements revealed several strong negative correlations (r < −0.7). These include correlations between UHML and SFI; UI with SFI and SFC(w); SFI with L(w) and L5%; UQL(w) with SFC(w) and SFC(n); Fine and IFC; and IFC and MatRatio. Additional significant negative correlations (r < −0.4) included UHML with MIC, FINE, and StdFine; MIC with UQL(w), L5%, and IFC; UI and SFC(n); STR and ELO; ELO with SFI, SFC(w), and SFC(n); SFI with L(n) and UQL(w); L(w) with SFC(w), SFC(n), and StdFine; UQL(w) with Fine and StdFine; and L5% with Fine and StdFine. Strong positive correlations (r > 0.7) were observed between UHML with L(w), UQL(w), and L5%; MIC and Fine; UI and L(n); L(w) with L(n), UQL(w), and L5%; L(n) and UQL(w); UQL(w) and L5%; SFC(w) and SFC(n); and Fine and MatRatio. Additionally, significant positive correlations (r > 0.4) were found between UHML with UI and L(n); MIC with MatRatio and StdFine; UI with ELO, L(w), and UQL(w); SFI with SFC(w) and SFC(n); L(n) with L5%; and Fine with StdFine.

Among the AFIS fiber quality traits, the strongest negative correlation was observed between IFC and MatRatio (r = −0.95). Strong negative correlations also existed between L(n) with both SFC(w) (r = −0.84) and SFC(n) (r = −0.77), as well as between Fine and IFC (r = −0.67). Significant but less strong negative correlations were also found between L(w) with SFC(w) (r = −0.54), SFC(n) (r = −0.41), and StdFine (r = −0.38); UQL(w) with Fine (r = −0.46) and StdFine (r = −0.53); and L5% with Fine (r = −0.49) and StdFine (r = −0.58). The strongest positive correlations were observed between UQL(w) and L5% (r = 0.98); SFC(w) and SFC(n) (r = 0.97); and L(w) and UQL(w) (r = 0.92). Additional strong positive correlations included L(w) and L(n) (r = 0.89); L(w) and L5% (r = 0.89); L(n) and UQL(w) (r = 0.65); and Fine and MatRatio (r = 0.69). There were also significant, though moderate, positive correlations between L(n) and L5% (r = 0.62), as well as for Fine with Std Fine (r = 0.59).

### 2.3. Effect of qFL-Chr.25 on HVI Fiber Traits Across Parental Genetic Backgrounds

To evaluate HVI fiber property traits, the NIILs were grouped by their four parental genetic backgrounds and compared to their recurrent parents (ACSJ4, DP50, GA089, and PMHS26) and the QTL donor parent, SL883 (Table 5). Across four years, SL883 had an average UHML of 32.9 mm, while the recurrent parents averaged 30.0 (ACSJ4), 29.5 (DP50), 30.8 (GA089), and 29.7 (PMHS26) mm. Several NIILs with the *qFL*-*Chr*.25 introgression (AC164Pos, DP213Pos, PM304Pos, PM321Pos) matched or exceeded SL883, with PM321Pos showing the highest UHML average of 33.4 mm. Across all eight sets of NIILs, QTL(+) lines consistently exhibited significantly longer fibers than QTL(−) lines. Figure 1 illustrates the UHML means with standard error and Waller-Duncan grouping letters for each genetic background, showing the close alignment of mean values, which underscores the QTL effects. The direct UHML differences between QTL(+) and QTL(−) NIILs are presented in Table 6, showing an average increase of 1.1 mm across all eight sets of NIILs due to *qFL*-*Chr*.25 introgression. The maximum and minimum difference between QTL(+) and QTL(−) lines were observed in DP213 and PM304 (1.9 mm) and GA437 (0.4 mm), respectively. Figure 2 depicts the scatterplots of the 16 near-isogenic introgression lines, with and without the QTL, grouped by their respective genetic backgrounds (Acala, Deltapine, Georgia, and Paymaster). These plots illustrate data across all replications and locations for the five different fiber length traits, including UHML, which showed significant differences across all four backgrounds.

Significant differences in micronaire (MIC) were observed across genotypes in each of the four parental backgrounds. The recurrent parents typically have coarser or more mature fiber compared to SL883, which had significantly finer or less mature fiber (Table 5). The NIILs showed an intermediate MIC value between SL883 and the recurrent parents, with SL883 averaging 3.7 while the recurrent parents ranged from 4.5 to 4.9. All NIILs fell between their two parents, except AC107Pos, which had a similar average MIC to SL883 at 3.7.

The *qFL*-*Chr*.25 QTL had little to no consistent effect on UI, SFI, ELO, or STR (Table 5). The UI ranged from 83.7 to 84.8%, with SL883 having the lowest uniformity while the recurrent parents were consistently above 84%. The SFI ranged from 5.1% to 6.9%, with SL883 averaging low at 5.3% and the recurrent parents generally above 6%. The ELO (elongation) percentages varied from 5.5% to 6.8%, with SL883 at 5.5% and the recurrent parents averaging 6% or higher. STR values ranged from 29.2 to 33.1 kN m kg^−1^, with SL883 averaging 31.9 kN m kg^−1^; the recurrent parents fell within the range mentioned earlier. Across all these traits, the NIILs varied among genetic backgrounds but averaged intermediate values compared to the parents, with the *qFL*-*Chr*.25 QTL showing little to no effect in most genetic backgrounds.

### 2.4. Effect of qFL-Chr.25 on AFIS Fiber Traits Across Parental Genetic Backgrounds

Table 7 presents AFIS fiber property traits for two NIIL sets across four genetic back- grounds: the background parents ACSJ4, DP50, GA089, and PMHS26, as well as the QTL donor parent SL883. For all AFIS length measurements, L(n), L(w), UQL(w), and L5%, the results showed a similar trend to the UHML trait, with SL883 consistently showing significantly longer fibers than the four background parents. However, not all eight sets of NIILs followed the UHML trend, where QTL(+) lines exhibited longer fibers than their QTL(−) counterparts. For the AFIS mean length by number measurement, L(n), the NIIL set AC107 in the ACSJ4 background, along with both sets of NIILs in the Georgia and PMHS26 backgrounds, did not show significant differences between the QTL(+) and QTL(−) lines (Figure 3). Additionally, for the mean length by weight measurement, L(w), the NIIL set AC164 in the ACSJ4 background, and both sets of NIILs in the Georgia background, did not show significant differences between the QTL(+) and QTL(−) lines (Figure 4). Finally, for both length measurements, UQL(w) (Figure 5) and L5% (Figure 6), which represent the longest 25% of fibers by weight and 5% span length by number in the samples, the NIIL set GA437 in the Georgia background did not show significant differences between the QTL(+) and QTL(−) lines. Across all genetic backgrounds, the average differences between QTL(+) and QTL(−) NIILs resulting from the *qFL*-*Chr*.25 introgression for L(n), L(w), UQL(w), and L5% were 0.6 mm, 0.1 mm, 1.1 mm, and 1.4 mm, respectively.

Figure 2 shows differences between QTL(+) and QTL(−) NIILs across the four backgrounds for AFIS fiber length traits. When analyzing the longer portion of fibers (UQL(w) and L5%), significant differences were observed between QTL(+) and QTL(−) lines in three of the four backgrounds, with the Georgia background being the exception. For the other two AFIS length traits, L(w) and L(n), a similar trend to Figure 3 and Figure 4 was observed, with the Deltapine background consistently showing significant differences among NIIL sets. A similar trend was observed in the Paymaster background, though significance was only detected for the L(w) trait.

Other AFIS measurements collected include short fiber content (SFC(w), SFC(n)), maturity (IFC, MatRatio), and fineness (Fine, StdFine). Short fiber content by weight (SFC(w)) and by number (SFC(n)) had variation from 4.2 to 6.2% (SFC(w)) and 12.9 to 18.1% (SFC(n)), respectively, with SL883 having higher percentages of short fiber content (5.9 and 17.9%) than the background parents. Among the NIILs, the QTL(+) lines generally exhibited higher short fiber content by both weight and number compared to the QTL(−) lines across all the genetic backgrounds, with the exception of the Deltapine background. For fiber maturity traits, the IFC varied between 4.5 and 6.0%, while the MatRatio ranged from 89 to 93%, with SL883 exhibiting a higher IFC and lower MatRatio compared to the background parents. Among the NIILs, only the QTL(+) lines in the Acala background were consistently showing higher fiber maturity compared to the QTL(−) lines. For fiber fineness, Sealand 883 had the finest fibers in the trial, with a fineness of 153.3 m/Tex, compared to the background parents, which had coarse fibers averaging 163.5 (ACSJ4), 177.6 (DP50), 170.1 (GA089), and 177.6 (PMHS26) m/Tex. Among the NIILs, the QTL(+) lines generally showed finer fibers than the QTL(−) lines across all genetic backgrounds, except the DP263 NIIL set, where the QTL(+) line had significantly coarser fibers. Finally, the ratio of fine to MatRatio is used to calculate standard fineness (StdFine), which provides an estimated value that closely correlates with fiber diameter and the resulting yarn strength [41]. The StdFine values ranged from 165.6 to 194.8 with SL883 and most of the QTL(+) lines showing lower values, possibly indicating biologically finer fiber compared to the QTL(−) lines and the background parents, except in the AC107 and DP263 sets.

## 3. Discussion

### 3.1. The Effect of qFL-Chr.25 on HVI UHML Fiber Length Across Parental Genetic Backgrounds

The interspecific introgression of genes, particularly those affecting fiber quality traits from extra-long staple cotton (*G. barbadense*), has long been a valuable strategy for improving upland cotton. The unique fiber properties of this species offer a novel source of fiber quality alleles for upland cotton, and germplasm lines possessing improved fiber properties derived from stable *G. barbadense* introgressions into upland cotton have been reported [9,20,22,30]. The development of TaqMan SNP genotyping markers (Table 1) has enabled quicker screening of large populations as well as the mining of valuable genetic resources outside of the primary gene pool of *G. hirsutum*. However, since fiber quality traits are quantitatively inherited, they are governed by many genes whose expression is sensitive to environmental influences. In addition, the genetic background into which fiber QTLs are introgressed can cause a tremendous amount of perturbation in their expression [42]. The interactions between fiber QTLs and genetic background are common in interspecific populations, even after several generations of backcrossing to the upland cotton parent [15,43,44]. Therefore, validating a fiber QTL across diverse genetic backgrounds and environments is essential to better understand the intricacies of these interactions before deploying it in breeding programs [30] or utilizing it for the cloning of fiber-related genes [33].

However, validating fiber quality QTLs, particularly for fiber length, is challenging due to their small phenotypic effects [45]. The use of marker-aided selection to develop near-isogenic introgression lines (NIILs) can reduce the “noise” in phenotypic data and minimize epistatic interactions, providing clearer insight into the specific effects of QTLs. By developing bulk sister lines with QTL(+) and QTL(−) NIILs across four genetic backgrounds, each represented by five to ten NIILs, Brown et al. [13] detected the effect of *qFL*-*Chr*.25 in only two of the four backgrounds. In a follow-up study, when individual QTL(+) and QTL(−) sister line pairs were evaluated, the QTL effect was evident in three of the four genetic backgrounds tested, while the Georgia background showed no statistical significance [30]. Across genetic backgrounds, QTL(+) NIILs had a mean UHML increase of 1.8 mm, with individual NIIL pairs exhibiting gains ranging from 0.6 mm to 3.5 mm over their corresponding QTL(−) sister lines.

The current study validated the phenotypic effect of *qFL*-*Chr*.25, showing that QTL(+) NIILs had significantly longer UHML compared to their QTL(−) sister lines across all genetic backgrounds and environments (Table 6). The average UHML increase was 1.1 mm, with gains ranging from 0.7 mm to 1.9 mm depending on the genetic background. These results confirm that *qFL*-*Chr*.25, which was introgressed from *G. barbadense* via Sealand883, has a positive effect on fiber length, reinforcing its potential value in breeding programs aimed at enhancing fiber length across diverse upland cotton germplasm.

### 3.2. Effect of qFL-Chr.25 on AFIS Length Measurements

The HVI measurement for fiber length, specifically UHML, does not capture the full range of fiber quality variation within a sample because it represents the average length of the longest 50% of the fibers by weight, potentially obscuring short fibers and over- representing longer fibers [46]. AFIS, in contrast, provides a precise characterization of individual fibers, offering a detailed analysis of the entire length profile and variations within a fiber sample [37]. It provides multiple length metrics, including the average fiber length weighted by number of fibers (L(n)), average fiber length weighted by fiber weight (L(w)), length of the longest 25% of fibers by weight (UQL(w)), and 5% span length by number of fibers (L5%) [47]. Hence, AFIS provides a more accurate prediction of spinning performance and yarn quality than HVI measurements [47]. However, since most cotton breeding programs evaluate anywhere from 2000 to 5000 fiber samples annually, HVI remains the preferred testing method due to AFIS’s significantly lower throughput and higher costs.

In this study, by comparing the AFIS fiber length distribution between the QTL(+) and QTL(−) NIILs, we observed notable variations in length distribution profiles attributed to the effect of the *G. barbadense* allele at *qFL*-*Chr*.25. As a natural material, individual cotton fibers vary considerably in length within a single sample. AFIS measurements that focus on the longest 5% (L5%) and 25% (UQL(w)) of fibers—much like UHML in HVI testing—showed that the QTL(+) NIILs produced longer fibers compared to the QTL(−) NIILs across all backgrounds, except Georgia (Figure 2). This lack of significance in the Georgia background is likely due to the GA437 NIIL set, which did not show significant differences between QTL(+) and QTL(−) lines, thereby masking the QTL effect within the background. It is noteworthy that this set of NIILs showed significant differences in the HVI fiber length data. Since the GA428 NIIL set did exhibit significant differences between QTL(+) and QTL(−) lines, this suggests that both NIIL sets must show significance for the QTL effect to be detectible within a given background. When looking at AFIS length measurements for the entire distribution of fibers, such as the average length weighted by fiber weight (L(w)) and average length weighted by number of fibers (L(n)), the effect of this QTL was not consistently observed. For example, among the eight NIIL pairs, only five showed a significant difference in L(w) (Figure 4), and only three showed a significant difference in L(n) (Figure 3). Collectively, these results suggest that the fiber length improvement conferred by *qFL*-*Chr*.25 primarily impacts the longer portion of fibers within a sample’s distribution.

It remains unclear whether the inconsistent impact of *qFL*-*Chr*.25 on L(w) and L(n) is due to biological factors, such as QTL expression, or artifacts introduced by these measurements. While both L(w) and L(n) are affected by the proportion of short fibers in a sample, L(n) is particularly sensitive, showing the lowest correlation among all fiber length measurements (Table 4), as short fibers contribute significantly to the fiber count but less to the sample’s overall weight. The QTL(+) NIILs produced significantly more short fibers across all backgrounds except in Deltapine, as indicated by higher short fiber content measurements (SFC(w) and SFC(n)), potentially lowering L(w) and L(n) measurements to levels comparable to those of the QTL(−) NIILs. Short fiber content, defined as the proportion of fibers shorter than 12.7 mm (1/2 length), may arise from an increased presence of immature fibers, which are prone to breakage during processing, including ginning and the mechanical sample preparation involved in AFIS measurement [48].

In future installments of this study, we will investigate whether the increased SFC observed in the QTL(+) NIILs arises from a wider spread in fiber length distributions (biological) or changes in other fiber traits, such as fineness and maturity. This is particularly relevant to the GA437 NIIL set in the Georgia background, which did not show significant length differences between QTL(+) and QTL(−) lines in AFIS measurements despite exhibiting significant differences in HVI fiber length. Both NIILs in this genetic background also exhibit significantly finer fibers and a higher proportion of short fibers. The increase in SFC, likely caused by broken fibers, could be influenced by differences in strength, maturity, or elasticity. A strong negative correlation exists between IFC and MatRatio (r = −0.95), suggesting that higher IFC may indicate lower maturity. However, further analyses, including cross-sectional and image analysis, are needed to confirm this and to determine whether fiber breakage skews IFC and SFC measurements. Additionally, longer fibers may contribute to increased IFC due to greater contact with saw ginning blades, thus causing more breakage.

### 3.3. Effect of qFL-Chr.25 on AFIS Fiber Fineness and Maturity Traits

After the cotton boll opens, the fibers are exposed to the environment, leading them to dry out. As a result, the lumen of individual fibers collapses, giving the fibers a bowed or kidney-shaped form [49]. Some fibers do not reach full maturity due to various factors. Immature fibers are undesirable because they are weaker, making them more prone to breakage during processing, and they also contain less cellulose, resulting in a lower capacity to absorb dye [50]. Unfortunately, micronaire values from HVI combine fiber fineness and maturity into a single reading instead of measuring each trait independently. AFIS provides separate measurements of fiber fineness (Fine) and maturity ratio (MatRatio). These direct measurements enable a more precise characterization of both fineness and maturity, avoiding the confounding effects inherent in the micronaire measurement [1].

The NIILs in this study are nearly isogenic rather than fully isogenic, making it plausible that other *Gb* introgressions from SL883 may contribute unique phenotypic differences not directly attributable to *qFL*-*Chr*.25. Kumar et al. [17] reported that SL883 retained at least 19 introgressed regions from the *Gb* parent. One potential off-target effect of these additional introgressions is consistently observed in fiber fineness measurements. Strong negative correlations were observed between fiber length traits focused on the longest portion of fibers (UHML, UQL(w), and L5%) and MIC, as well as Fine and StdFine (r = −0.49 and above). The negative correlation indicates that longer fibers tend to be finer, as lower fineness values correspond to finer fibers, while higher fineness values are associated with coarse fibers for both AFIS and HVI measurements. For example, with the exception of the DP263 sets of NIILs, all NIILs showed micronaire values following a similar trend to the fiber length traits, with the QTL(+) lines trending toward SL883, which has finer fibers than the recurrent parents, and the QTL(−), exhibiting coarser fibers similar to the recurrent parents. Similarly, the QTL(+) lines have lower Fine values than the QTL(−) in all genetic backgrounds except in the DP263 set of NIILs. Standard fineness, which provides an estimate of the fiber diameter without the bias of maturity, also followed this trend, except in the NIIL sets AC107 and DP263. Another interesting observation was that some of the QTL(−) NIILs showed an increase in fiber length compared to their recurrent parents. Plausible explanations for these observations could be the presence of other genetic regions introgressed from SL883 that were not specifically targeted during the NIILs development or that the *qFL*-*Chr*.25 region may also influence fiber fineness, which warrants further investigation. Finer mapping of the QTL region is needed to identify the genetic regions contributing to both the fiber fineness and length traits.

## 4. Materials and Methods

### 4.1. Plant Materials

The cotton lines utilized in this study were developed by the University of Georgia’s (UGA) Cotton Molecular Breeding Laboratory (CMBL) from crossing Sealand 883 (SL883) (PI 528875) with four regionally adapted parents, Acala SJ4 (ACSJ4) (PI 529538), Deltapine 50 (DP50) (PVP, 8400154; PI 529566), Paymaster HS26 (PMHS26) (PVP, 8600087; PI 606814), and GA 2004089 (GA089). The development of the plant materials has been described by Brown et al. [30] and is outlined in Figure 7. Briefly, for each SL883 × genetic background crosses, two sets of near-isogenic introgression sister lines (NIILs) were developed. In each set, one NIIL was homozygous for the SL883 allele at *qFL*-*Chr*.25, representing the QTL(+) lines, and the other lacking the SL883 allele at *qFL*-*Chr*.25, representing the QTL(−) line. These NIILs were generated by bulking seeds from single *F*_4_ plants derived from *F*_3_ plants heterozygous for the *qFL*-*Chr*.25 region, as defined by the flanking SSR markers NAU2713 and CIR109. In 2013, 15 individual *F*_5_ plants tracing back to different *F*_3:4_ plants from each genetic background cross were screened for the *qFL*-*Chr*.25 QTL in the greenhouse. Two sets of QTL(+) and QTL(−) NIILs were selected from each genetic background, resulting in a total of 16 NIILs for the study. Among these, three sets (DP213, GA437, PM321) were newly developed, while five sets had also been tested by Brown et al. [30] (AC107, AC164, DP263, GA428, PM304). In 2019, the remnant *F*_3:5_ seeds from each NIIL were genotyped using novel TaqMan SNP genotyping assays to validate their genotype prior to this study (Table 1).

### 4.2. Field Experiments

In 2020 and 2023, the 16 NIILs, along with the QTL donor parent and the four background parental lines, were planted at the UGA’s Gibbs Farm, Tifton, GA in a randomized complete block design with three replications. The soil types encountered in the testing area at Gibbs Farm in Tifton, GA include Tifton loamy sands, a fine-loamy, kaolinitic, ther- mic Plinthic Kandiudult; and Clarendon loamy sands, a fine-loamy, siliceous, semiactive, thermic Plinthaquic Paledult. The plots were two rows wide, spaced 1 m apart, and 12.2 m long. In 2020, a 75-boll sample was hand-harvested from the mid-fruiting zone of each plot to obtain sufficient seed for line perpetuation and fiber analysis.

In 2021 and 2022, the trial was again planted at UGA Gibbs Farm along with two additional locations in a randomized complete block design with four replications instead of three. Multiple replications across eight environments helped normalize the effects of *qFL*-*Chr*.25, providing sufficient statistical power to detect the QTL of interest. The locations include the University of Georgia’s Southwest Georgia Research Station in Plains, GA, and Texas A&M University’s Research Farm in College Station, TX. The soil types found at UGA’s Southwest Georgia Research Station in Plains, GA, include a Greenville sandy loam—a fine, kaolinitic, thermic Rhodic Kandiudult; Faceville sandy loam—a fine kolinitic thermic Typic Kandiudults; and Tifton sandy loams. The soil type encountered in College Station, TX, was a Westwood silt loam—a fine-silty, mixed thermic Fluventic Ustochrept—integrated with a Ships clay—a fine, mixed, thermic Udic Chromustert.

### 4.3. Fiber Quality Measurement

A 50-boll sample was hand-harvested from each plot for both High Volume Instrument (HVI) and Advanced Fiber Information System (AFIS) fiber testing. After harvest, samples were ginned on a laboratory 10-saw gin, and fiber was sent to the Fiber Quality Laboratory at Cotton Incorporated in Cary, NC. The lab conducted HVI and AFIS testing on each sample. Fiber parameters measured by HVI included upper half mean length (UHML), micronaire (MIC), uniformity index (UI), strength (STR), elongation (ELO), and short fiber index (SFI) [51]. The fiber parameters measured by AFIS included mean length by weight (L(w)), upper quartile length (UQL(w)), short fiber content by weight (SFC(w)), mean length by number (L(n)), 5% span length by number (L5%), short fiber content by number (SFC(n)), fineness (Fine), immature fiber content (IFC), and maturity ratio (MatRatio) [41]. An additional measurement is referred to as standard fineness (StdFine), which is the ratio of Fine to MatRatio, and it gives an estimate of fiber diameter [52].

### 4.4. Statistical Analysis

The statistical analyses were performed using SAS Enterprise Guide, Version 8.3 Update 7 [53], and JMP 17 Pro [54]. In SAS, the PROC GLM function was used for the analysis of variance (ANOVA) and mean separations using Waller-Duncan’s method to test for significant differences in fiber quality traits between genotypes. Fiber quality data were analyzed separately for each location. The homogeneity of variances across locations was tested on the error mean squares, and the data were combined (Table 2). For data combined across multiple years and locations, years and replications were treated as random effects, while entry and location were considered fixed effects. A probability threshold of 0.05 was applied to determine significant differences among trait means. JMP 17 Pro [54] was used for the Pearson correlation analysis of all fiber quality traits tested in this study. JMP averaged the lines across replicates and locations after verification of consistency for the trait of interest across environments.

## 5. Conclusions

In this study, we validated the potential phenotypic gain on fiber length from deploy- ing *qFL*-*Chr*.25 derived from interspecific introgression. While the effect of *qFL*-*Chr*.25 may seem modest, averaged increases in fiber length measured by UHML, L(n), L(w), UQL(w), and L5% were 1.1 mm, 0.6 mm, 0.1 mm, 1.1 mm, and 1.4 mm, respectively, across all genetic backgrounds. This level of phenotypic gain represents a meaningful advancement in the context of cotton fiber quality improvement. To put this in perspective, achieving a mm increase in length through incremental improvements in the US cotton industry took approximately 20 years [55]. The QTL(+) NIILs carrying the *qFL*-*Chr*.25 allele from *G. barbadense* have been publicly released; hence, this QTL could be immediately deployed in commercial breeding programs. Finally, the diagnostic markers flanking this QTL have been converted into a TaqMan SNP genotyping platform (Table 1), enabling automated, reproducible, and scalable screening of large populations typical in commercial breeding programs, thereby enhancing its utility and making it highly valuable for the continued improvement of fiber quality as it competes with synthetic fibers.

## Figures and Tables

**Figure 1 plants-14-01937-f001:**
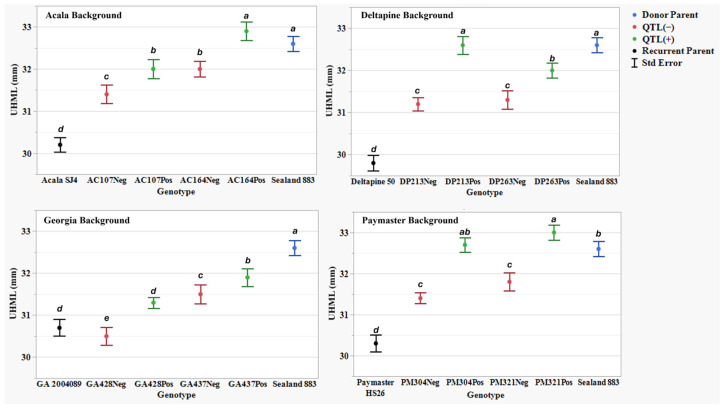
Upper half mean length (UHML) (mm) means, standard errors, and mean separations (k = 100, based on Waller-Duncan LSD, represented with alphabet letters) for sixteen near-isogenic introgression lines differing for *qFL*-*Chr*.25, deployed within four regionally adapted genetic backgrounds (Acala, Deltapine, Georgia, and Paymaster) compared to the recurrent cultivar parents (ACSJ4, DP50, GA089, PMHS26) and the QTL donor parent (SL883), in Tifton, GA, in 2020 and 2023, College Station, TX, Plains, GA, and Tifton, GA, in 2021 and 2022.

**Figure 2 plants-14-01937-f002:**
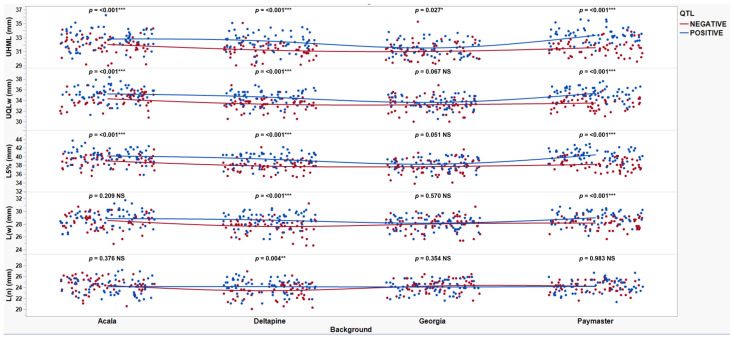
Scatterplots of near-isogenic introgression lines with QTL(+) (blue) and QTL(−) (red) by background (Acala, Deltapine, Georgia, and Paymaster) for all fiber length traits (UHML, UQL(w), L5%, L(w), and L(n)) in mm tested in Tifton, GA, in 2020 and 2023, College Station, TX, Plains, GA, and Tifton, GA, in 2021 and 2022. *, **, and *** represent significance with *p*-values of 0.05, 0.01, and 0.001, respectively. NS denotes not significant.

**Figure 3 plants-14-01937-f003:**
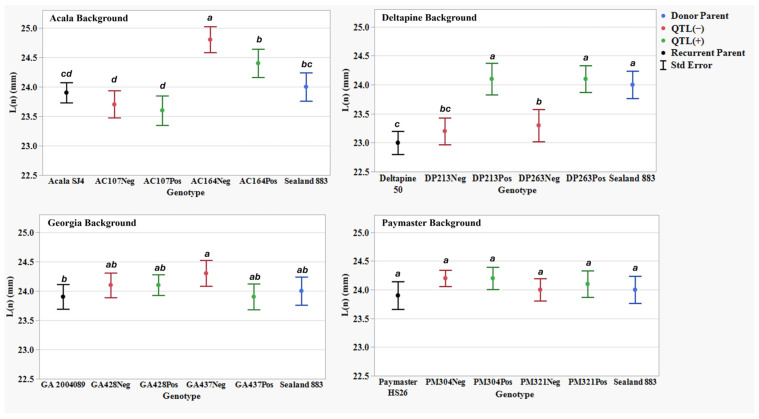
Length by number L(n) (mm) means, standard errors, and mean separations (k = 100, based on Waller-Duncan LSD, represented with alphabet letters) for sixteen nearly isogenic introgression lines differing for *qFL*-*Chr*.25, deployed within four regionally adapted genetic backgrounds (Acala, Deltapine, Georgia, and Paymaster) compared to the recurrent cultivar parents (ACSJ4, DP50, GA089, PMHS26) and the QTL donor parent (SL883), in Tifton, GA, in 2020 and 2023, College Station, TX, Plains, GA, and Tifton, GA, in 2021 and 2022.

**Figure 4 plants-14-01937-f004:**
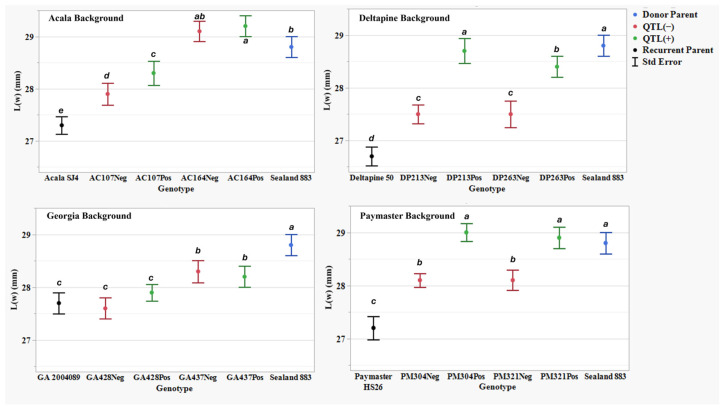
Length by weight L(w) (mm) means, standard errors, and mean separations (k = 100, based on Waller-Duncan LSD, represented with alphabet letters) for sixteen nearly isogenic introgression lines differing for *qFL*-*Chr*.25, deployed within four regionally adapted genetic backgrounds (Acala, Deltapine, Georgia, and Paymaster) compared to the recurrent cultivar parents (ACSJ4, DP50, GA089, PMHS26) and the QTL donor parent (SL883), in Tifton, GA, in 2020 and 2023, College Station, TX, Plains, GA, and Tifton, GA, in 2021 and 2022.

**Figure 5 plants-14-01937-f005:**
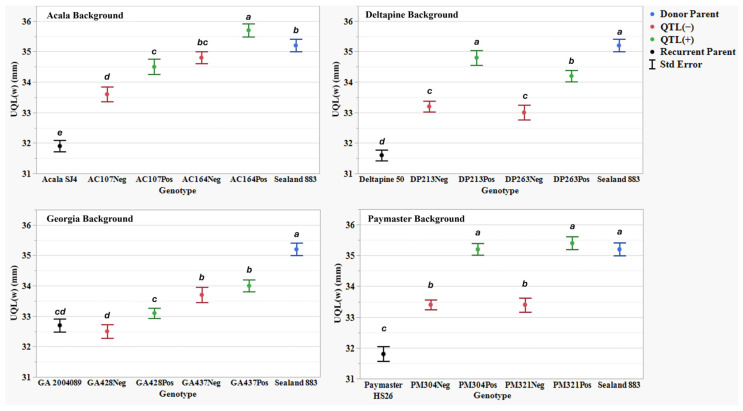
Upper quartile length (UQL(w)) (mm) means, standard errors, and mean separations (k = 100, based on Waller-Duncan LSD, represented with alphabet letters) for sixteen nearly isogenic introgression lines differing for *qFL*-*Chr*.25, deployed within four regionally adapted genetic backgrounds (Acala, Deltapine, Georgia, and Paymaster) compared to the recurrent cultivar parents (ACSJ4, DP50, GA089, PMHS26) and the QTL donor parent (SL883), in Tifton, GA, in 2020 and 2023, College Station, TX, Plains, GA, and Tifton, GA, in 2021 and 2022.

**Figure 6 plants-14-01937-f006:**
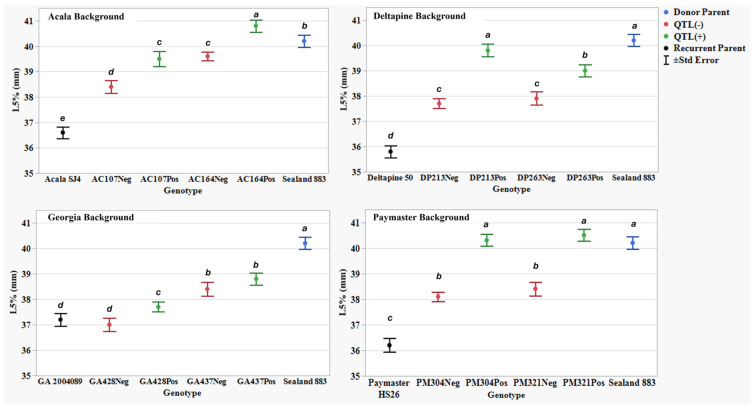
The 5% span length by number (L5%) (mm) means, standard errors, and mean separations (k = 100, based on Waller-Duncan LSD, represented with alphabet letters) for sixteen nearly isogenic introgression lines differing for *qFL*-*Chr*.25, deployed within four regionally adapted genetic backgrounds (Acala, Deltapine, Georgia, and Paymaster) compared to the recurrent cultivar parents (ACSJ4, DP50, GA089, PMHS26) and the QTL donor parent (SL883), in Tifton, GA, in 2020 and 2023, College Station, TX, Plains, GA, and Tifton, GA, in 2021 and 2022.

**Figure 7 plants-14-01937-f007:**
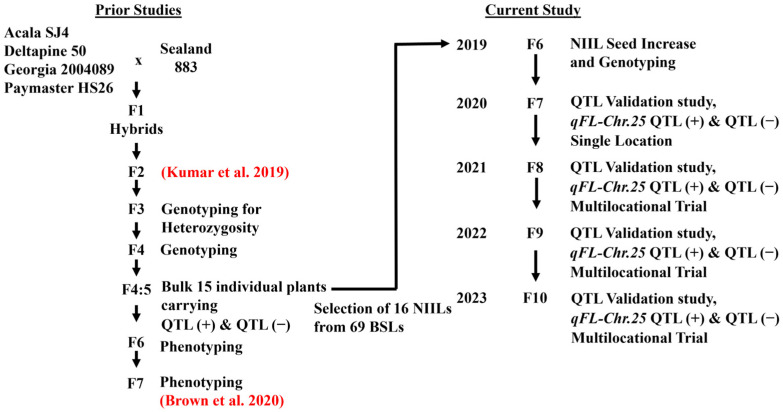
Flowchart for population development including original cross, previous studies [17,30], and current study.

**Table 1 plants-14-01937-t001:** Attributes of SNP markers associated with *qFL*-*Chr*.25.

	SNP Marker *^a^*	
	i17091Gh *^b^*	i10579Gh
Chromosome	D06	D06
SNP location *^c^*	867,431	1,072,116
Forward primer	TCTTTCTCTTTCCTCCCCTCTCTAC	TACTCATTCGTTGGTTCGTCAAGT
Reverse primer	GGACATGGCTGGTTGAAGTGA	GGTGGCCTCGAAAATCAGAA
*G. barbadense* allele	C	T
*G. hirsutum* allele	T	C
“Gb” allele probe *^d^*	AATAGCTCCTCCCCCAT	TTTTCCTCCCGGTCG
“Gh” allele probe	TAGCTCCTCCTCCATC	TCCTCCCGGCCGC
50 bp sequence spanning the SNP marker	CTACCTCTCCAACAATAGCTCCTCC[C/T]CCATCTTCACTGCACCGTCACTTCA	TGGATACTCCTCGTTTTCCTCCCGG[T/C]CGCCGCCCTTCTGATTTTCGAGGCC

^*a*^ SNP information from 63K SNP Array [39], retrieved from data deposited at CottonGen (https://www.cottongen.org). ^*b*^ SNP site is different from originally reported. ^*c*^ Genome coordinates from *G. hirsutum* ‘TM-1’ reference [40]. ^*d*^ ‘Gb’ allele probe is associated with introgressed chromatin from *G. barbadense*, and ‘Gh’ allele probe is associated with *G. hirsutum* chromatin.

**Table 2 plants-14-01937-t002:** Analysis of variance mean squares for HVI traits across eight environments.

Source	DF	^†^ UHML	MIC	UI	STR	ELO	SFI
Rep	3	2.6	0.12	1.3	3.4	0.05	1.0
Env	7	48.8 **	3.62 **	106.2 **	61.3 **	44.81 **	139.9 **
Error A	19	1.7 **	0.14 **	1.4	3.3 *	0.22 **	0.7 *
Genotype	20	40.5 **	3.55 **	2.4 **	21.7 **	4.63 **	7.8 **
GxE	140	0.6	0.09 **	0.8	2.1 *	0.15 **	0.6 **
Error B	440	0.8	0.05	0.8	1.5	0.07	0.3

* Significant at the 0.01 probability level. ** Significant at the 0.001 probability level. ^†^ UHML, upper half mean length; MIC, micronaire; UI, uniformity index; STR, fiber bundle strength; ELO, fiber elongation; SFI, short fiber index.

**Table 3 plants-14-01937-t003:** Analysis of variance mean squares for AFIS traits across eight environments.

Source	DF	^†^ L(w)	L(n)	UQL(w)	L5%	SFC(w)	SFC(n)	Fine	IFC	MatRatio	StdFine
Rep	3	1.6	2.6	3.0	5.0	3.8	22	19	1.0	0.0009	29
Env	7	61.0 **	82.8 **	55.4 **	75.1 **	84.1 **	491 **	3000 **	147.1 **	0.1827 **	586 **
Error A	19	1.3 **	1.3 **	1.5 *	1.9 *	1.4 **	7 *	89 **	0.8 **	0.0006 **	56 **
Genotype	20	13.7 **	4.7 **	43.1 **	62.7 **	10.6 **	86 **	1508 **	4.4 **	0.0054 **	1497 **
GxE	140	0.6	0.7*	0.8	1.0	0.9 **	5 **	30	0.4 **	0.0003 *	19 *
Error B	440	0.5	0.5	0.7	0.9	0.5	3	23	0.2	0.0002	13

* Significant at the 0.01 probability level. ** Significant at the 0.001 probability level. ^†^ L(w), length by weight; L(n), length by number; UQL(w), upper quartile length by weight; L5%, 5% span length by number; SFC(w), short fiber content by weight; SFC(n), short fiber content by number; Fine, fineness; IFC, immature fiber content; MatRatio, maturity ratio; StdFine, standard fineness.

**Table 4 plants-14-01937-t004:** Pearson correlation coefficients among High Volume Instrument (HVI, in blue) and Advanced Fiber Information System (AFIS, in green) fiber quality traits across eight environments.

	HVI Traits						AFIS Traits								
Trait	UHML ^†^	MIC	UI	STR	ELO	SFI	L(w)	L(n)	UQL(w)	L5%	SFC(w)	SFC(n)	Fine	IFC	MatRatio
MIC	−0.55 **														
UI	0.35 **	−0.01													
STR	0.15 **	−0.01	−0.03												
ELO	0.01	0.01	0.56 **	−0.43 **											
SFI	−0.67 **	0.34 **	−0.68 **	0.06	−0.50 **										
L(w)	0.8 **	−0.32 **	0.55 **	0.13 *	0.19 **	−0.69 **									
L(n)	0.54 **	−0.07	0.67 **	0.03	0.36 **	−0.64 **	0.89 **								
UQL(w)	0.9 **	−0.49 **	0.35 **	0.16 **	−0.01	−0.63 **	0.92 **	0.65 **							
L5%	0.9 **	−0.53 **	0.31 **	0.13 **	0.01	−0.66 **	0.89 **	0.62 **	0.98 **						
SFC(w)	−0.11 *	−0.24 **	−0.65 **	0.03	−0.49 **	0.48 **	−0.54 **	−0.84 **	−0.2 **	−0.17 **					
SFC(n)	0.01	−0.26 **	−0.58 **	0.11 *	−0.50 **	0.40 **	−0.41 **	−0.77 **	−0.06	−0.05	0.97 **				
Fine	−0.55 **	0.81 **	0.02	−0.07	0.05	0.21 **	−0.31 **	−0.11 *	−0.46 **	−0.49 **	−0.23 **	−0.22 **			
IFC	0.21 **	−0.42 **	0.02	−0.09	0.14 **	0.08	0.06	0.04	0.11 *	0.08	0.18 **	0.12 *	−0.67 **		
MatRatio	−0.17 **	0.43 **	−0.01	0.15 **	−0.19 **	−0.05	−0.03	−0.03	−0.08	−0.07	−0.18 **	−0.10	0.69 **	−0.95 **	
StdFine	−0.55 **	0.62 **	0.04	−0.27 **	0.28 **	0.33 **	−0.38 **	−0.12 *	−0.53 **	−0.58 **	−0.10 *	−0.19 **	0.59 **	0.15 **	−0.18 **

* and ** represent significance with *p*-values of 0.01 and 0.001, respectively. ^†^ UHML, upper half mean length; MIC, micronaire; UI, uniformity index; STR, strength; ELO, fiber elongation; SFI, short fiber index; L(w), length by weight; L(n), length by number; UQL(w), upper quartile length by weight; L5%, 5% span length by number; SFC(w), short fiber content by weight; SFC(n), short fiber content by number; Fine, fineness; IFC, immature fiber content; MatRatio, maturity ratio; StdFine, standard fineness.

**Table 5 plants-14-01937-t005:** High volume instrument (HVI) fiber property means of NIILS differing for *qFL*-*Chr*.25 within four regionally adapted backgrounds compared to the recurrent and QTL donor parents across eight environments.

Background	Genotype	UHML ^†^	MIC	UI	STR	ELO	SFI
		mm	unit	%	kN m kg^−1^	%	%
Acala	ACSJ4	30.0 d ^‡^	4.5 a	84.2 b	32.7 ab	6.5 a	6.3 a
	AC107Neg	31.6 c	4.2 b	84.0 bc	32.4 abc	5.8 b	6.1 a
	AC107Pos	32.4 b	3.7 d	83.7 c	31.8 c	5.7 b	5.7 b
	AC164Neg	32.4 b	4.0 c	84.8 a	33.1 a	5.6 c	5.5 bc
	AC164Pos	33.2 a	3.8 d	84.1 bc	32.7 a	5.2 d	5.3 c
	SL883	32.9 a	3.7 d	83.7 bc	31.9 bc	5.5 c	5.3 c
Deltapine	DP50	29.5 d	4.8 a	84.2 a	29.2 c	6.8 a	6.9 a
	DP213Neg	31.1 c	4.2 b	84.2 a	31.2 b	6.0 c	6.6 b
	DP213Pos	33.0 a	3.9 d	84.2 a	32.5 a	5.6 d	5.3 d
	DP263Neg	31.3 c	3.9 d	83.9 a	30.8 b	5.9 c	6.3 b
	DP263Pos	32.0 b	4.0 c	84.3 a	32.1 a	6.4 b	5.7 c
	SL883	32.9 a	3.7 e	83.7 a	31.9 a	5.5 d	5.3 d
Georgia	GA089	30.8 d	4.7 a	84.56 a	30.4 b	6.5 a	6.3 ab
	GA428Neg	30.5 e	4.0 c	84.3 ab	31.5 a	6.3 b	6.3 a
	GA428Pos	31.0 d	3.9 d	84.6 a	31.7 a	6.1 c	6.1 ab
	GA437Neg	31.6 c	4.2 b	84.1 bc	31.9 a	6.1 c	6.0 bc
	GA437Pos	32.0 b	4.0 cd	84.1 bc	31.4 a	5.9 d	5.8 c
	SL883	32.9 a	3.7 e	83.7 c	31.9 a	5.5 e	5.3 d
Paymaster	PMHS26	29.7 d	4.9 a	84.6 a	31.9 a	6.0 a	6.3 a
	PM304Neg	31.4 c	4.3 b	84.5 a	32.1 a	5.6 cd	6.0 b
	PM304Pos	33.3 ab	4.0 c	84.0 b	32.2 a	5.6 c	5.1 cd
	PM321Neg	31.7 c	4.1 c	84.2 ab	31.6 a	5.9 b	6.0 b
	PM321Pos	33.4 a	3.9 d	83.9 b	31.8 a	5.5 de	5.0 d
	SL883	32.9 b	3.7 e	83.7 b	31.9 a	5.5 e	5.3 c
CV (%)		3.5	7.8	1.7	4.9	13.2	24.0

**^†^** UHML, upper half mean length; MIC, micronaire; UI, uniformity index; STR, fiber bundle strength; ELO, fiber elongation; SFI, short fiber index. ^‡^ Alphabet letters next to means represent significance for Waller-Duncan LSD (k = 100) within each background.

**Table 6 plants-14-01937-t006:** Upper half mean length (UHML) improvement attributed to the presence of *qFL*-*Chr*.25 within eight NIILs deployed in four different backgrounds across eight environments.

Background	NIILs	QTL(+)	std dev	QTL(−)	std dev	Difference
Acala	AC107	32.4	1.3	31.6	1.2	0.8
	AC164	33.2	1.2	32.4	1.0	0.8
Deltapine	DP213	33.0	1.2	31.1	0.9	1.9
	DP263	32.0	1.0	31.3	1.2	0.7
Georgia	GA428	31.0	0.7	30.5	1.7	0.5
	GA437	32.0	1.1	31.6	1.2	0.4
Paymaster	PM304	33.3	1.0	31.4	0.7	1.9
	PM321	33.4	1.0	31.7	1.2	1.7
					Average:	1.1

**Table 7 plants-14-01937-t007:** Advanced Fiber Information System (AFIS) fiber property means of NIILS differing for *qFL*-*Chr*.25 within four regionally adapted backgrounds compared to the recurrent and QTL donor parents across eight environments.

Background	Genotype	L(w) ^†^	L(n)	UQL(w)	L5%	SFC(w)	SFC(n)	Fine	IFC	MatRatio	StdFine
		mm	mm	mm	mm	%	%	m/Tex	%		Hs
Acala	ACSJ4	27.3 e ^‡^	23.9 cd	32.0 e	36.6 e	4.3 d	13.0 d	163.5 a	4.9 d	0.93 a	176.1 a
	AC107Neg	28.0 d	23.8 d	33.7 d	38.4 d	5.6 b	16.6 b	157.6 bc	5.2 c	0.93 a	169.9 c
	AC107Pos	28.4 c	23.8 d	34.6 c	39.5 c	6.1 a	18.0 a	154.2 de	5.8 a	0.90 c	171.3 bc
	AC164Neg	29.2 ab	24.9 a	34.9 bc	39.6 c	4.8 c	14.9 c	159.5 b	5.2 c	0.93 a	171.9 b
	AC164Pos	29.3 a	24.5 b	35.8 a	40.8 a	5.8 ab	17.7 a	155.7 cd	5.5 b	0.92 b	169.7 cd
	SL883	28.9 b	24.1 bc	35.2 b	40.2 b	5.9 ab	17.9 a	153.3 e	5.6 b	0.91 b	168.2 d
Deltapine	DP50	26.8 d	23.1 c	31.7 d	35.8 d	5.6 b	16.3 d	177.6 a	5.3 d	0.91 ab	194.8 a
	DP213Neg	27.6 c	23.3 bc	33.3 c	37.7 c	6.2 a	18.1 a	162.2 b	5.6 b	0.91 b	178.7 b
	DP213Pos	28.8 a	24.2 a	34.9 a	39.8 a	5.8 ab	17.6 abc	155.8 c	5.4 cd	0.92 a	170.3 d
	DP263Neg	27.6 c	23.5 b	33.2 c	37.9 c	5.9 ab	17.0 bcd	152.3 d	5.9 a	0.90 c	169.4 de
	DP263Pos	28.4 b	24.1 a	34.2 b	39.0 b	5.5 b	16.7 cd	157.3 c	5.9 a	0.90 c	175.0 c
	SL883	28.9 a	24.1 a	35.2 a	40.2 a	5.9 ab	17.9 ab	153.3 d	5.6 bc	0.91 ab	168.2 e
Georgia	GA089	27.7 c	23.9 b	32.8 cd	37.2 d	5.0 c	15.2 c	170.1 a	5.1 c	0.92 a	184.6 a
	GA428Neg	27.7 c	24.1 ab	32.5 d	37.0 d	4.5 d	13.5 e	160.6 b	6.0 a	0.89 c	179.7 b
	GA428Pos	27.9 c	24.1 ab	33.1 c	37.7 c	4.7 cd	14.5 d	156.2 c	5.9 a	0.90 c	174.2 c
	GA437Neg	28.3 b	24.4 a	33.8 b	38.4 b	4.9 c	15.0 cd	162.5 b	5.6 b	0.91 b	178.6 b
	GA437Pos	28.3 b	24.0 ab	34.1 b	38.8 b	5.5 b	16.5 b	157.8 c	5.5 b	0.91 b	173.3 c
	SL883	28.9 a	24.1 ab	35.2 a	40.2 a	5.9 a	17.9 a	153.3 d	5.6 b	0.91 b	168.2 d
Paymaster	PMHS26	27.4 c	24.1 a	31.9 c	36.2 c	4.2 d	12.9 d	177.6 a	4.5 d	0.95 a	187.4 a
	PM304Neg	28.1 b	24.3 a	33.4 b	38.1 b	4.7 c	14.4 c	161.0 b	5.0 c	0.93 b	172.9 b
	PM304Pos	29.0 a	24.2 a	35.2 a	40.3 a	5.7 a	17.8 a	156.8 c	5.0 c	0.93 b	169.0 c
	PM321Neg	28.2 b	24.1 a	33.6 b	38.4 b	5.0 b	15.4 b	157.4 c	5.3 b	0.92 c	171.5 b
	PM321Pos	29.0 a	24.2 a	35.4 a	40.5 a	5.8 a	17.9 a	152.1 d	5.2 b	0.92 c	165.6 d
	SL883	28.9 a	24.1 a	35.2 a	40.2 a	5.9 a	17.9 a	153.3 d	5.6 a	0.91 c	168.2 c
CV (%)		2.4	3.0	2.5	2.5	13.5	10.8	3.0	9.0	1.6	2.0

**^†^** L(w), length by weight; L(n), length by number; UQL(w), upper quartile length by weight; L5%, 5% span length by number; SFC(w), short fiber content by weight; SFC(n), short fiber content by number; Fine, fineness; IFC, immature fiber content; MatRatio, maturity ratio; StdFine, standard fineness. ^‡^ Alphabet letters next to means represent significance for Waller-Duncan LSD (k = 100) within each background.

## Data Availability

The original data presented in the study are openly available in CottonGen: The Community Database for Cotton Genomics, Genetic and Breeding Research at https://www.cottongen.org/sites/default/files/cottongen_submitted/Wan_et_al_2025.xlsx.

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
