# Peer review of "Phenotypic Validation of the Cotton Fiber Length QTL, qFL-Chr.25, and Its Impact on AFIS Fiber Quality"

_plants, 2025, doi:10.3390/plants14131937_

Round 1

Reviewer 1 Report

Comments and Suggestions for Authors

This manuscript focused on validating the potential phenotypic gain on fiber length from developing qFL-Chr.25 derived from interspecific introgression, and investigating the effect of the QTL introgression in different lines. Generally, we believe this manuscript are of innovation and significance for cotton studies, while many technical issues must be put forward first.

  • In the Abstract, too many background descriptions were observed, and suggest to add more results in this portions;
  • Most of tables in this manuscript had the long titles, and were not the three-line format. In the table 2, we noticed the gibberish. Meanwhile, suggest to add the note information for explaining the meanings of a-d in Table 4 and 5.
  • TaqMan SNP genotyping platform was only observed in the Discussion and Conclusions, and suggest to add the relative description in the results.
  • In the references, gene names and Latin names were not italic, and lake of the issue number.
Comments on the Quality of English Language

The English could be improved to more clearly express the research.

Author Response

  1. Comment: In the Abstract, too many background descriptions were observed, and suggest to add more results in this portions

Response: We have revised the Abstract to be more concise and within the 200 words limit.

  1. Most of the tables in the manuscript have long titles, and were not the three-line format. In the table 2, we noticed the gibberish. Meanwhile, suggest to add the note information for explaining the meanings of a-d in Table 4 and 5.

Response: we have revised all Figure and Table titles.

  1. TaqMan SNP genotyping platform was only observed in the Discussion and Conclusion, and suggest add the relative description in the results

Response: To provide additional context on the SNP assay, we added the following sentences: in the Results section (line 508), we stated, “TaqMan SNP genotyping markers (Table 1) were able to identify G. barbadense specific alleles and allowed for the prescreening of material prior to field evaluations.” In the Discussion section (line 576), we added, “The development of TaqMan SNP genotyping markers (Table 1) has enabled quicker screening of large populations as well as the mining of valuable genetic resources outside of the primary gene pool of G. hirsutum.”).

  1. Comment: “In the references, gene names and latin were not italic, and lake of issue number”

Response: fixed.

Reviewer 2 Report

Comments and Suggestions for Authors

1、Why does the environmental effect have a greater impact than the genotype, especially in traits like UI, STR, ELO, and SFI? The impact on UHML is also significantly higher than that of the genotype. Can it be considered that qFL-Chr.25 is affected by the environment? If the environment changes, can this QTL still be detected? Additionally, the same is true for the detection results of AFIS traits, and the value of Fine even reaches 3000. Should we consider issues related to phenotypic measurements?

2、It is recommended to add a comparison of the accuracy between Advanced Fiber Information System (AFIS) and HVI measurements.

3、The discussion in the article is not in - depth enough. It is only limited to the effects of this QTL and does not describe its applications and production value.

4、The layout of the titles in the first column of Table 2 has text overlapping issues.

5、All the figures in the article are very unclear, which seriously affects reading.

Author Response

  1. Comment: “Why does the environmental effects have a greater impact than the genotype, especially in traits like UI, STR, ELO, and SFI? The impact on UHML is also significantly higher than that of the genotype. Can it be considered that qFL-Chr.25 is affected by the environment? If the environment changes, can this QTL still be detected? Additionally, the same is true for the detection results of AFIS traits, and the value of Fine even reaches 3000. Should we consider issues related to phenotypic measurements?”

Response: Fiber quality traits of interest to breeders are known to have a complex genetic architecture and are highly influenced by environmental factors. This complexity is evident in the variability of trait expression across environments—some traits being more affected than others. As a result, many fiber QTLs reported in the literature fail to reproduce consistently across genetic backgrounds and environments. Our study represents a rare case in which a fiber length QTL has been validated across multiple genetic backgrounds and over diverse environments. The use of near-isogenic lines enabled us to statistically demonstrate that this QTL is consistently expressed across different genetic backgrounds (GA, PM, DP, and Acala), geographical locations (Georgia and Texas), and growing seasons (2020, 2021, 2022, and 2023). We believe this provides compelling evidence that the QTL is stable and will serve as a valuable resource for future cotton fiber improvement.

  1. Comment: “It is recommended to add a comparison of the accuracy between AFIS and HVI measurements”

Response: This concern was also noted by an additional Reviewer. As clarified, our study was designed primarily to validate the presence and effect of the QTL using the Advanced Fiber Information System (AFIS), rather than to perform a direct comparison between phenotyping platforms. Notably, our findings indicate that HVI, which emphasizes measurements of the longest portion of the fiber sample, consistently detected the QTL across all genetic backgrounds. In contrast, AFIS failed to detect the QTL in certain backgrounds. However, it's important to highlight that while HVI effectively captures key length-related parameters, it does not provide detailed insights into the full fiber length distribution, short fiber content, or fiber maturity—traits that are better characterized by AFIS. This highlights the complementary nature of both phenotyping platforms, each offering unique and valuable perspectives on fiber quality traits.

  1. Comment: “The discussion in the article is not in-depth enough. It is only limited to the effects of the QTL and does not describe its applications and production value”

Response: The applications and production value of the QTL was addressed in the Conclusions section. We underscore the significance of our findings by noting that “achieving a mm increase in length through incremental improvements in the U.S. cotton industry took approximately 20 years”. Furthermore, we highlight the practical utility of this QTL by stating that “diagnostic markers flanking for this QTL have been converted into a TaqMan SNP genotyping platform (Table 1), enabling automated, reproduceable, and scalable screening of large populations typical in commercial breeding programs, thereby enhancing its utility and making it highly valuable for the continued improvement of fiber quality as it competes with synthetic fibers.”

  1. Comment: “The layout of the titles in the first column of Table 2 has text overlapping issues”

Response: Table 2 (now Table 3): has been adjusted

  1. Comment: “All the figures in the article are very unclear, which seriously affects reading” Response: Figures have been adjusted to help readability

Round 2

Reviewer 2 Report

Comments and Suggestions for Authors

corrections to minor methodological errors and text editing,and English could be improved to more clearly express the research.

Author Response

Reviewer 2, Round 2 Response:

1. Comment "The English could be improved to more clearly express the research."

Response: Changes made and highlighted

2. “corrections to minor methodological errors and text editing, and English could be improved to more clearly express the research.”

Response: Changes made and highlighted